# Do All Critically Ill Patients with COVID-19 Disease Benefit from Adding Tocilizumab to Glucocorticoids? A Retrospective Cohort Study

**DOI:** 10.3390/v15020294

**Published:** 2023-01-20

**Authors:** Cristina Mussini, Alessandro Cozzi-Lepri, Marianna Meschiari, Erica Franceschini, Giulia Burastero, Matteo Faltoni, Giacomo Franceschi, Vittorio Iadisernia, Sara Volpi, Andrea Dessilani, Licia Gozzi, Jacopo Conti, Martina Del Monte, Jovana Milic, Vanni Borghi, Roberto Tonelli, Lucio Brugioni, Elisa Romagnoli, Antonello Pietrangelo, Elena Corradini, Massimo Girardis, Stefano Busani, Andrea Cossarizza, Enrico Clini, Giovanni Guaraldi

**Affiliations:** 1Department of Infectious Diseases, Azienda Ospedaliero-Universitaria Policlinico of Modena, 41124 Modena, Italy; 2Department of Surgical, Medical, Dental and Morphological Sciences University of Modena and Reggio Emilia, 41124 Modena, Italy; 3Centre for Clinical Research, Epidemiology, Modelling and Evaluation (CREME), Institute for Global Health, UCL, London, WC1E 6BT, UK; 4Clinical and Experimental Medicine PhD Program, Department of Biomedical and Metabolic Sciences and Neurosciences, University of Modena and Reggio Emilia, 41124 Modena, Italy; 5Respiratory Diseases Unit, Azienda Ospedaliero-Universitaria Policlinico of Modena, 41124 Modena, Italy; 6Internal Medicine Department, Azienda Ospedaliero-Universitaria Policlinico of Modena, 41124 Modena, Italy; 7Department of Medical and Surgical Sciences for Children and Adults, University of Modena and Reggio Emilia, 41124 Modena, Italy; 8Department of Anaesthesia and Intensive Care Unit, Azienda Ospedaliero-Universitaria Policlinico of Modena, 41124 Modena, Italy

**Keywords:** COVID-19, tocilizumab, glucocorticoids

## Abstract

Background: Treatment guidelines recommend the tocilizumab use in patients with a CRP of >7.5 mg/dL. We aimed to estimate the causal effect of glucocorticoids + tocilizumab on mortality overall and after stratification for PaO_2_/FiO_2_ ratio and CRP levels. Methods: This was an observational cohort study of patients with severe COVID-19 pneumonia. The primary endpoint was day 28 mortality. Survival analysis was conducted to estimate the conditional and average causal effect of glucocorticoids + tocilizumab vs. glucocorticoids alone using Kaplan–Meier curves and Cox regression models with a time-varying variable for the intervention. The hypothesis of the existence of effect measure modification by CRP and PaO_2_/FiO_2_ ratio was tested by including an interaction term in the model. Results: In total, 992 patients, median age 69 years, 72.9% males, 597 (60.2%) treated with monotherapy, and 395 (31.8%), adding tocilizumab upon respiratory deterioration, were included. At BL, the two groups differed for median values of CRP (6 vs. 7 mg/dL; *p* < 0.001) and PaO_2_/FiO_2_ ratio (276 vs. 235 mmHg; *p* < 0.001). In the unadjusted analysis, the mortality was similar in the two groups, but after adjustment for key confounders, a significant effect of glucocorticoids + tocilizumab was observed (adjusted hazard ratio (aHR) = 0.59, 95% CI: 0.38–0.90). Although the study was not powered to detect interactions (*p* = 0.41), there was a signal for glucocorticoids + tocilizumab to have a larger effect in subsets, especially participants with high levels of CRP at intensification. Conclusions: Our data confirm that glucocorticoids + tocilizumab vs. glucocorticoids alone confers a survival benefit only in patients with a CRP > 7.5 mg/dL prior to treatment initiation and the largest effect for a CRP > 15 mg/dL. Large randomized studies are needed to establish an exact cut-off for clinical use.

## 1. Introduction

After more than one year since the beginning of the COVID-19 pandemic and the advent of vaccines, we are still facing a relevant number of cases among those not vaccinated or vaccine non-responders [1]. Unfortunately, in-hospital mortality is still over 20%, and a few drugs are shown to be effective in decreasing mortality in patients with severe pneumonia enrolled in randomized clinical trials. The first large randomized study to show a benefit in the reduction of mortality was the RECOVERY trial demonstrating the efficacy of dexamethasone [2]. Indeed, its positive results in patients needing oxygen supplementation changed the standard of care dramatically, and all patients with a moderate or severe clinical picture at hospital admission who require oxygen are now treated with glucocorticoids. Nevertheless, the 28-day mortality in patients receiving dexamethasone alone in the RECOVERY trial remained high (22%), leading clinicians to seek additional therapeutic strategies in an attempt to improve the survival rate.

Since the beginning of the epidemic, tocilizumab, an IL-6 antagonist [3], was considered a good candidate for treating the inflammatory phase of the disease, the so-called cytokine storm [4], which plays a role in the development of acute respiratory distress syndrome, thromboembolic disease, acute kidney injury, and vasculitis [5], all known complications of the disease. The infection of type II pneumocytes in the lungs may induce the accumulation of inflammatory cells consisting of neutrophils, macrophages, and T-lymphocytes with a massive production of various cytokines [6]. In particular, IL-6 may also mediate the activation of endothelial cells that produce pro-inflammatory cytokines and can contribute to the onset of coagulopathy [6,7].

At the beginning of the epidemic, observational studies [8,9], including some from our group [10], showed promising results of tocilizumab, although early results from randomized clinical trials were conflicting [11,12,13,14,15]. Our study showed higher effectiveness of tocilizumab when compared to standard of care, in particular in patients with severe gas exchange impairment, measured with PaO_2_/FiO_2_ ratio [9].

Recently, a randomized trial conducted on critically ill patients showed a survival benefit of tocilizumab and sarilumab vs. standard of care [16]. As a consequence, a few treatment guidelines (Italy and UK) recommended the use of immune-modulators for the treatment of severe COVID-19 pneumonia [17,18]. More recently, the RECOVERY consortium has presented the results of the comparison between patients randomized to receive dexamethasone (monotherapy) or dexamethasone plus tocilizumab (add-on therapy). The trial showed a significant effect of adding tocilizumab to standard of care in reducing in-hospital mortality, regardless of the stage of the COVID-19 disease at hospital admission [19], determining a change also in the recommendations included in the NIH guidelines [20]. The inclusion criterion for entering the RECOVERY trial on tocilizumab was a level of C-reactive protein (CRP) > 7.5 mg/dL and hypoxia, defined as oxygen saturation <92% on room air; thus, all patients requiring oxygen and with this level of CRP should receive glucocorticoids and tocilizumab. However, this threshold was arbitrary, and more recently, the CORIMUNO-TOCI trial performed a post-hoc analysis and suggested that it should be increased to 15 mg/dL [21]. 

The aim of this analysis was two-fold. First to use real-world data to replicate the results of the RECOVERY trial, which showed a benefit of therapy with glucocorticoids + tocilizumab vs. glucocorticoids alone, especially in patients with CRP > 7.5 mg/dL. Second, to investigate whether the difference in risk of death between the two strategies varies according to the most recent levels of gas exchange impairment and/or CRP prior to treatment initiation in order to inform treatment guidelines regarding the best timing of initiation. 

## 2. Methods

This was a retrospective, observational cohort study carried out at the University Hospital of Modena, Italy. We included all patients attending the facility who were admitted to the hospital because of severe COVID-19 pneumonia over the period June 2020- June 2021. None of the participants were admitted for conditions other than COVID pneumonia or developed COVID-19 after entering the hospital. Data were obtained from electronic health records and complied fully with Italian law on personal data protection and the ethics committee of the Area Vasta Emilia Romagna Nord, who approved the study (396/2020/OSS/AOUMO – Cov-2 MO-Study). The retrospective data were fully anonymized, and the only sensitive data were the year of birth. All consecutive adult patients (≥18 years) with severe COVID-19 pneumonia, as previously defined, enrolled after June 16, 2020 were included in this analysis [22]. 

### 2.1. Procedures

We aimed to use this observational cohort to emulate the RECOVERY trial design. The treatment strategies evaluated were: treatment with glucocorticoids + tocilizumab vs. glucocorticoids alone. Glucocorticoids included dexamethasone or methylprednisolone. Dexamethasone was used according to standard of care at 6 mg/day for 10 days. Methylprednisolone 2 mg/kg body weight/day was initiated in patients admitted to the intensive care unit (ICU) for treatment of ARDS [23,24]. In addition, patients were treated with low molecular weight heparin at a prophylactic dose soon after entering the hospital as part of the standard of care [25]. 

The glucocorticoids + tocilizumab group included a non-random subset of participants in whom tocilizumab was added to glucocorticoid monotherapy upon respiratory deterioration. Tocilizumab was administered intravenously at 8 mg/kg bodyweight (up to a maximum of 800 mg) twice, 12 h apart [10]. Respiratory deterioration was defined by clinical judgment upon detection of >30% reduction of PaO_2_/FiO_2_ ratio within two consecutive blood gas analyses within 24 h and/or onset of tachypnea defined as >25 breaths per minute (bpm) or new onset of respiratory distress defined as subjective breathing discomfort and use of the accessory respiratory muscles. Participants in whom the use of tocilizumab was contraindicated were excluded [10]. Exclusion criteria for tocilizumab use were described elsewhere [10]. All patients provided oral informed consent for treatment with tocilizumab.

### 2.2. Outcome Measures

The primary outcome of the study was a 28-day all-cause mortality rate. The distribution of causes of death by treatment group was also described. Secondary outcomes were the incidence of major adverse events such as neutropenia, severe liver disease, sepsis, pulmonary embolism, and thrombosis over follow-up. These events were graded according to the Common Terminology Criteria for Adverse Events (CTCAE) and included severe and life-threatening adverse events (grade of severity 3–4) [26].

### 2.3. Statistical Analysis

Baseline characteristics of the participants, assessed at the time of hospitalization, were compared after stratification by treatment strategy started in follow-up (glucocorticoids + tocilizumab vs. glucocorticoids alone). Continuous variables were expressed as median (IQR) and compared by the Mann–Whitney U test. Categorical variables were expressed as numbers and percentages and compared by χ^2^ test or Fisher’s exact test by treatment strategy. 

To estimate the effect of glucocorticoids + tocilizumab on the risk of day 28 death, we used a weighted pooled logistic regression model to approximate the parameters of a marginal structural Cox model by mean of inverse probability weights with the aim to emulate the RECOVERY trial [2,27]. Day 28 in-hospital death was used because this was the primary endpoint in the reference trial. Participants’ follow-up accrued from the date of hospital admission until death or the date of discharge. Administrative censoring on 30 June 2021 was also applied. Weights have been calculated using the predicted values from the pooled logistic models for the probabilities of glucocorticoids + tocilizumab and those of censoring, respectively. Treatment with glucocorticoids + tocilizumab and glucocorticoids alone were fitted as time-dependent interventions. According to our assumptions, age, ethnicity, duration of symptoms, baseline (at hospital admission) PaO_2_/FiO_2_ ratio, CRP, and Charlson comorbidity index (CCI) were identified as the main time-fixed confounders of our comparison of interest. In addition, we also created the weights to control for the following time-varying potential confounders: the most recent value of PaO_2_/FiO_2_ ratio and CRP prior to treatment initiation, post-baseline use of remdesivir, and invasive mechanical ventilation. We also performed a number of sensitivity analyses: (i) after excluding participants who used high-dose steroids in follow-up, (ii) after excluding participants with solid organ cancer or age >75 years who were less frequently candidates for invasive mechanical ventilation (IMV) because of poor prognosis, and (iii) after excluding participants who had received a full vaccination cycle prior to hospital admission. 

Unweighted Kaplan–Meier estimates of the day 28 cumulative risk of death accounting for competing risk were calculated. Analysis was repeated after weighting for baseline confounders. In these analyses, baseline was the time to initiate each of the interventions, and participants who recovered and were discharged alive before day 28 were given a time of follow-up of 28 days. The majority of participants were sent home, while for those who were discharged to nursing homes or other departments in the hospital or other institutions, alive status at day 28 was individually checked. In contrast, in the marginal structural model analysis, potential informative censoring was controlled for using inverse probability of censoring weights. Weighted hazard ratios (HRs) with 95% confidence intervals (CI) were shown together with the unadjusted and standard adjustment for time-dependent covariates. 

We then used a standard multivariable Cox regression analysis and repeated the main comparison between glucocorticoids + tocilizumab vs. glucocorticoids alone across a number of subsets after stratifying participants by level of most recent PaO_2_/FiO_2_ ratio and CRP prior to time of treatment initiation. Specifically, as the cut-off for the stratification, for the PaO_2_/FiO_2_ ratio, we used the Q1, median, and Q3 quartiles of the distribution of the most recent value prior to treatment initiation. For CRP, instead, we used a priori clinical cut-offs of 2, 7.5 (corresponding to current guidelines), and 15 mg/dL. Interactions between the intervention and study population strata were formally tested by including a multiplicative term in the Cox regression model. Age-adjusted HRs with 95% CI in the strata from fitting a standard Cox regression model were shown in a forest plot. A two-sided test of less than 0.05 was considered statistically significant. All statistical analyses were performed using the SAS software, version 9.4 (Carey, Cary, NC, USA).

## 3. Results

### 3.1. Study Population

Over the study period, a total of 992 patients with severe COVID-19 pneumonia were admitted to our hospital. All patients were treated with glucocorticoids, and 395 (39.8%) received glucocorticoids + tocilizumab over follow-up. The standard dosage of glucocorticoids was used in 902 participants (90.9%), 565 (94.6%) in the glucocorticoids alone group vs. 337 (85.3%) in the glucocorticoids + tocilizumab group. The remaining 90 participants (9.1%; 32 in glucocorticoids alone vs. 58 in glucocorticoids + tocilizumab) were treated with high-dose methylprednisolone 2 mg/kg/day. Methylprednisolone was more frequently used in the glucocorticoids + tocilizumab group (14.7% vs. 5.4%, *p* < 0.001; Appendix A). Overall, 98% of participants started treatment within 14 days of the date of hospital admission. 

The epidemiological characteristics of the study population are shown in Table 1. Overall, 368 (37.1%) were females, the median age was 69 years (range 57–80), and 95.2% were of Caucasian ethnicity. The two treatment groups were balanced concerning sex, age, ethnicity, and median duration of symptoms. In contrast, the prevalence of comorbidities at hospital admission was higher in the glucocorticoids alone group, including solid cancer (41% vs. 31%, *p* < 0.001), cerebrovascular disease (27% vs. 17%, *p* < 0.001), chronic kidney failure (28% vs. 21%, *p* = 0.017), liver failure, (22% vs. 15%, *p* = 0.002) and dementia (29% vs. 19%; *p* < 0.001). Overall, there was no difference in the age-unadjusted CCI (*p* = 0.38). Vice versa, gas exchange impairment at hospitalization was significantly lower in patients who were treated with glucocorticoids + tocilizumab vs. glucocorticoids alone: median PaO_2_/FiO_2_ ratio was 235 vs. 276 mmHg (*p* < 0.001). The proportion of patients with PaO_2_/FiO_2_ ≤250 mmHg was 59% vs. 36% (*p* < 0.001), and the proportion of patients with PaO_2_/FiO_2_ ≤150 mmHg was 24% vs. 12% (*p* < 0.001) (Table 2). Participants in the glucocorticoids + tocilizumab group also had higher median respiratory rate: 24 vs. 20 bpm (*p* < 0.001). Median values of the laboratory parameters are shown in Table 3: patients receiving glucocorticoids + tocilizumab had a higher baseline level of IL-6, CRP, and lactate dehydrogenase (LDH), while total lymphocyte count was significantly lower. The median value of CRP at baseline was overall 6.0 mg/dL (IQR 3.0, 15.0) and, although statistically significant, was not clinically different in the 2 groups: 6.0 mg/dL (IQR 2.0, 14.0) in glucocorticoids alone vs. 7.0 mg/dL (IQR 4.0, 15.0) (*p* < 0.001) in those treated with glucocorticoids + tocilizumab. In summary, at time of hospital admission, the group of patients who were treated with glucocorticoids alone were more likely to have concomitant comorbidities while those treated with glucocorticoids + tocilizumab showed on average greater inflammatory state and worse gas exchange impairment.

### 3.2. Concomitant Treatment

Concerning additional treatments, the two strategies did not differ with respect to heparin use which was provided at prophylactic dose, intermediate, and full dose in 58.8%, 33.6%, and 2.9% of patients in both arms, while remdesivir was used more frequently in patients receiving glucocorticoids + tocilizumab (3.0 % vs. 0.8%; *p* = 0.009, Appendix A). The criteria for invasive or non-invasive respiratory support were similar in the two strategies, as shown by median PaO_2_/FiO_2_ ratios collected on the day of starting different respiratory supports (Table 2). 

Nevertheless, high flow nasal oxygen (HFNO) therapy (51% vs. 12%, *p*< 0.001), continuous positive airway pressure (C-PAP)/non-invasive mechanical ventilation (NIV) (28 % vs. 6%; *p* < 0.001), and IMV (13% vs. 6%; *p* < 0.001) were more frequently used in patients receiving glucocorticoids + tocilizumab (Appendix A). None of the participants included were already receiving respiratory support at hospital admission. Importantly, some of the factors described in Table 1, Table 2 and Table 3 are time-varying so that the levels observed at baseline for this analysis (i.e., initiation of treatment) may differ from those measured at hospital admission. Nevertheless, the multivariable analyses correctly control for the values at baseline using time-varying covariates.

### 3.3. Replication of the Results of the Recovery Trial

In the unweighted competing risk Kaplan–Meier analysis, by 28 days from treatment initiation, the proportion who died was 16.2% (95% CI: 13.3–19.2%) in the glucocorticoids alone group vs. 15.7% (95% CI: 12.1–19.2%) in the glucocorticoids + tocilizumab group (Figure 1A, log-rank *p* = 0.51). Overall, 69% of participants were discharged before day 28, less frequently in the glucocorticoids + tocilizumab group (63% vs. 74%, *p* < 0.0001), and overall, the majority (74%) were sent home (Appendix A). Of note, small differences between the curves, especially after day 18, are likely to reflect the baseline status of participants who intensified who were likely to have experienced a deterioration in respiratory function as well as a worsening in the level of inflammation. Indeed, after controlling for baseline confounders, there was greater evidence for a difference in mortality between the two treatment groups (log-rank *p* = 0.25, Figure 1B). Furthermore, when we evaluated the effect of glucocorticoids + tocilizumab after controlling for baseline and post-baseline confounders in the Cox regression model, the difference in risk between the intervention groups was in favor of the glucocorticoids + tocilizumab strategy and remained significant in the marginal structural model estimates (weighted HR (wHR) = 0.59 95% CI:0.38–0.90, *p* = 0.015, Table 4). Results were similar after removing participants who initiated high doses of glucocorticoids over follow-up (wHR = 0.59, 95% CI: 0.38–0.93, *p* = 0.022), and the effect of glucocorticoids + tocilizumab was only slightly attenuated after removing those aged 75 + or with a diagnosis of solid tumor (wHR = 0.64, 95% CI: 0.38–1.07, *p* = 0.087, Appendix A, respectively). Interestingly, in the subset of 681 participants who received a full vaccination cycle prior to hospital admission, we only observed 3 deaths (0.4%). After restricting the analysis to participants who were not vaccinated, the results were again similar (weighted model HR = 0.64, 95% CI (0.42–0.98, *p* = 0.04, Appendix A).

### 3.4. Subset Analyses

When we evaluated the effect of the glucocorticoids + tocilizumab in subsets of the population in the standard Cox model, we found significant evidence that the most recent PaO_2_/FiO_2_ prior to treatment initiation was an effect measure modifier for the glucocorticoids + tocilizumab strategy with similar RH across the strata (*p* = 0.02, Figure 2). In particular, the effect of glucocorticoids + tocilizumab vs. glucocorticoids alone was the largest in participants with a PaO_2_/FiO_2_ < 150 mmHg. 

Similarly, for CRP, we detected a dose-response effect of tocilizumab with glucocorticoids + tocilizumab showing little effect in participants with a CRP of 2.1–7.5 and increased benefit for levels of CRP> 7.5 mg/dL with the largest effect seen for those starting with a CRP > 15 mg/dL. Interestingly, intensification with tocilizumab appeared to have a large effect also in participants who initiated with very low levels of CRP (in the <2 mg/dL range). However, there was only one event in the tocilizumab + glucocorticoids group, and the estimate is very imprecise. Additionally, the formal test for interaction for CRP yielded a non-significant result (*p* = 0.57, Figure 2).

### 3.5. Adverse Events

There were some differences in the prevalence of causes of death by treatment group (chi-square *p* = 0.0007). The prevalence of death due to non-COVID-19-associated pre-existing conditions was higher in the glucocorticoids alone group (29/113, 25.7%) vs. the glucocorticoids + tocilizumab therapy group (5/79, 6.3%) while the opposite tendency was observed for deaths due to respiratory insufficiency (25/113, 22.1% vs. 30/79, 38.0%), respectively (Appendix A). We also evaluated the incidence of adverse events by treatment groups. Participants receiving glucocorticoids + tocilizumab showed a higher incidence of neutropenia (5.8% vs. 2.3%, *p* = 0.02) and severe liver disease (18.0% vs. 10.3%, *p* = 0.003), while we found no evidence for a difference in the incidence of other conditions such as pulmonary embolism (6.1% vs. 3.7%, *p* = 0.08), sepsis (8.8% vs. 8.5%, *p* = 0.08), or thrombosis (1.3% vs. 0.3%, *p* = 0.12, Appendix A). 

## 4. Discussion

Our findings confirm the important role of glucocorticoids plus tocilizumab in patients with critical and severe COVID-19 pneumonia. Our analysis shows that response to tocilizumab is enhanced in the presence of a high level of inflammation. Although tocilizumab is an IL-6 antagonist, this cytokine is rarely measured in the clinics; therefore, the level of inflammation is often monitored using CRP. There is an ongoing debate regarding whether CRP should be used to guide treatment initiation with tocilizumab and, if so, at which exact level to start glucocorticoids + tocilizumab. The RECOVERY trial showed that in participants with oxygen saturation <92% on room air or receiving oxygen therapy and a CRP ≥7.5 mg/dL, glucocorticoids + tocilizumab provided a benefit regardless of all other participants’ characteristics at entry in the trial [2]. IDSA guidelines were tailored to the inclusion criteria of this trial [28]. However, more recently, a post-hoc analysis of the CORIMUNO-TOCI French trial suggested that this threshold should be elevated to 15 mg/dL, and our larger study, even if not randomized and underpowered to detect such an interaction, also suggests that CRP might be an important effect measure modifier [21]. Specifically, we suggest that further trials should be conducted to identify the best clinical cut-off to be used in the clinic. Further studies are needed to evaluate whether delaying the initiation of glucocorticoids + tocilizumab until CRP reaches a level >15 mg/dL could improve the benefit provided by this treatment, avoiding the unnecessary onset of adverse events of a drug that also has limited availability. Our findings also support a recent hypothesis by Ascierto et al., according to whom the maximum benefit of tocilizumab can be achieved if it is administered during a finite time window, after the outset of hyper-inflammation but before the process of tissue damage becomes irreversible [29]. 

Our results, even if from an observational cohort, are robust since the estimated effect of glucocorticoids + tocilizumab was comparable to that observed in the RECOVERY trial [2]. Unfortunately, the much smaller sample size of our study implies wider uncertainty around our estimate. Of note, the inclusion criteria in the trial were broader than in our study, in which tocilizumab was always administered as a double infusion regardless of deterioration or improvement of blood gas exchange 12 h after the first dose. 

We believe that our results are very relevant and contribute to identifying patients who could benefit the most from glucocorticoids + tocilizumab. Indeed, although the use of tocilizumab plus glucocorticoids is now recommended by international guidelines [29], there is still an ongoing debate regarding whether CRP is the most important and only marker to be used to guide initiation of tocilizumab in patients failing glucocorticoids and which clinical cut-off should be used. Earlier analysis from our group showed that the effect of tocilizumab was greater at low values of PaO_2_/FiO_2_ ratio, which was confirmed here, with participants with a PaO_2_/FiO_2_ < 150 mmHg at the time of initiation glucocorticoids + tocilizumab being those showing the greatest benefit [10]. Tocilizumab is an IL-6 receptor antagonist, and a plasma level of IL-6 >30 pg/mL has also been associated with both severe COVID-19 and response to treatment [30,31,32]. Unfortunately, we could not stratify the analysis by IL-6 levels because too many patients had missing values for this marker. Specific protocol studies are needed to investigate the utility of IL-6 for guiding therapy initiation in this setting. 

Our study suffers from a few limitations. Firstly, it is not a randomized study, and therefore, it relies on a correct specification of the model and no residual or unmeasured confounding. This may include unreported decisions to withhold treatment related to poor prognosis (e.g., palliative intent), which may stand to reason why patients who did not benefit from tocilizumab would experience the highest mortality. However, results were similar after excluding participants who were less likely to be candidates for IMV. We observed an excess of deaths due to preexisting comorbidities in the glucocorticoids alone group. However, measurement error is always a possibility in the observational setting where causes of death or the presence of comorbidities are determined without a chart review or screening.

Confounding by indication was also strong as patients intensified with tocilizumab if they were failing glucocorticoids. However, confounding bias was minimized by the use of inverse probability weights. In addition, the fact that participants treated with tocilizumab were those who had previously failed glucocorticoids should have introduced a conservative bias in favor of the glucocorticoids alone group. The effect of adjustment was particularly clear when comparing the unweighted and weighted KM curves, the latter showing a much bigger difference between the treatment groups, especially after day 18. Also importantly, statistical power to detect effect measure modification was low due to the small sample size and number of events, especially in the group with CRP of 0–2 mg/dL. Contrary to our hypothesis, the effect of intensification was larger in the CRP of 0–2 mg/dL range group vs. the 2.1–7.5 range group, and estimates from the model were very imprecise. This latter finding could also simply reflect the direct effect of low inflammation on the risk of developing the outcome, which is not fully controlled by the regression adjustment. Nevertheless, and despite the slightly different inclusion criteria in the two studies, after our propensity score adjustment, our estimates of the treatment effect were consistent with those observed in the RECOVERY trial. Moreover, the benefit of glucocorticoids + tocilizumab was larger in our analysis than in the trial, and this could be due to effect measure modification by CRP, consistent use of two doses, or unmeasured confounding bias. High-dose glucocorticoids were more frequently used in the glucocorticoids + tocilizumab group, but results were similar in a sensitivity analysis excluding participants who used them. 

Participants exposed to both glucocorticoids and tocilizumab could develop infections due to induced immunosuppression. Indeed, in the univariable analysis, we found an increased risk of neutropenia and severe liver disease in the glucocorticoids + tocilizumab group vs. glucocorticoids alone, although absolute clinical risk remained below 20%. 

Finally, survival bias is also an important source of distortion, as participants had to survive until the date of starting one of the treatment strategies to be included in this analysis. However, because of the characteristics of our hospital, for approximately 60% of participants, the date of starting treatment coincided with the day of hospital admission, and it has been shown that for a time window of misclassification shorter than 5 days, immortal-time bias is negligible [33]. Importantly, 95% of the participants were Caucasians, so results cannot be generalized to patients of other ethnicities, and the fact that all were enrolled in the same hospital prevents the generalization of the results to other settings. Of note, the vast majority of severe COVID-19 disease and deaths were observed in participants who had not received full vaccination prior to hospital admission. As such, the results of our analysis are less applicable to patients currently admitted to hospital whose risk of severe disease and death is largely reduced by vaccination and, more recently, circulating variants.

On the other hand, our study also has many strengths. First, the research topic is ubiquitous for obvious reasons. To our knowledge, ours is the largest study conducted so far, including patients treated with tocilizumab in a real-life hospital setting, and no randomized trial powered to detect effect measure modification by CRP exists or has been planned. Secondly, the data were extremely rich, with a complete report of demographics, comorbid conditions, and laboratory and blood gas data. Laboratory and blood gas variables are key confounding factors that were collected prospectively in a standardized way, and participants were followed up for a minimum of 28 days. A sophisticated method of analysis was employed to estimate, under a set of assumptions, the average and conditional causal effects of treatment. Importantly, this led to an estimate of the treatment effect, which was consistent with that coming from randomized studies. In addition, the linkage between electronic charts of blood counts and clinical data allowed us to perform in-depth stratified analyses after grouping by most recent levels of CRP and gas exchange impairment prior to treatment initiation. 

In conclusion, this analysis contributes to the search for better identifying patients who could benefit the most from glucocorticoids + tocilizumab as compared to glucocorticoids alone in terms of reduction in mortality. Specifically, our data are consistent with current evidence suggesting that glucocorticoids + tocilizumab in patients with a CRP of 2.1–7.5 mg/dL at the time of treatment initiation has limited effect, while it provides substantial survival benefit in those with CRP of 7.5–15 and even larger if >15 mg/dL. Large randomized studies are needed to establish an exact cut-off for clinical use.

## Figures and Tables

**Figure 1 viruses-15-00294-f001:**
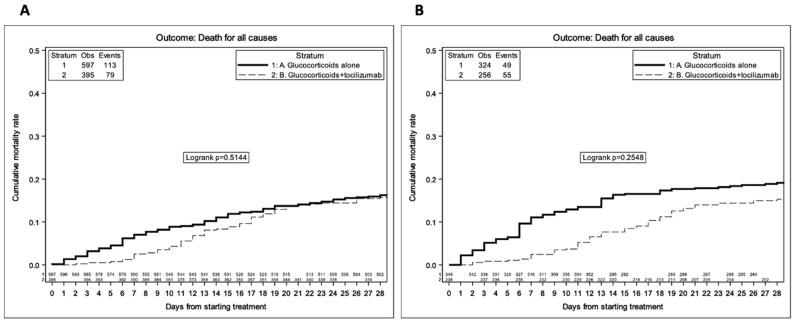
Survival KM estimates stratified by treatment group—unweighted analysis (**A**) and weighted analysis after adjustment for age, ethnicity, CCI, baseline CRP, and PaO_2_/FiO_2_ ratio (**B**).

**Figure 2 viruses-15-00294-f002:**
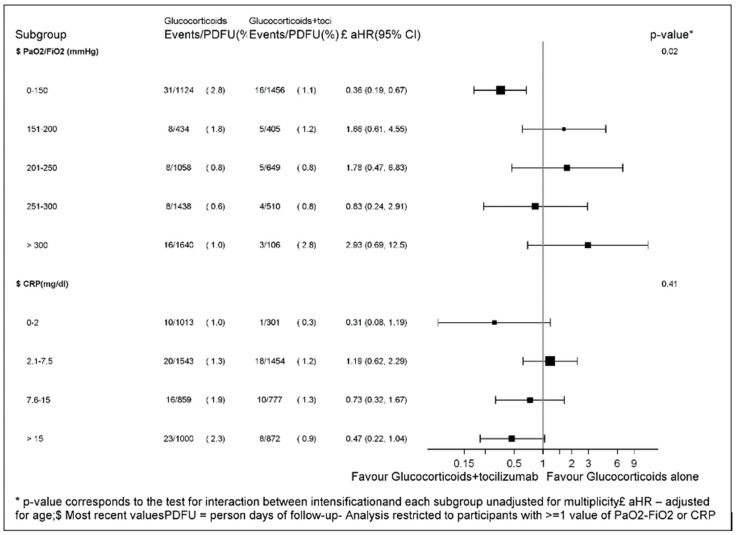
Forest plot of the effect of glucocorticoids + tocilizumab vs. glucocorticoids alone across baseline PaO_2_/FiO_2_ and CRP strata.

**Table 1 viruses-15-00294-t001:** Demographics, comorbidities and main delays by treatment group.

	Treatment Strategy			
Characteristics	Glucocorticoids	Glucocorticoids + Tocilizumab	*p*-Value *	Total
	N = 597	N = 395		N = 992
Age, years			0.428	
Median (IQR)	70 (56, 81)	69 (60, 78)		69 (57, 80)
Gender, *n*(%)			0.543	
Female	226 (37.9%)	142 (35.9%)		368 (37.1%)
Ethnicity, *n*(%)			0.540	
Caucasian	564 (94.5%)	380 (96.2%)		944 (95.2%)
Black	10 (1.7%)	3 (0.8%)		13 (1.3%)
Asian	18 (3.0%)	10 (2.5%)		28 (2.8%)
Hispanic	5 (0.8%)	2 (0.5%)		7 (0.7%)
Comorbidities, *n*(%)				
>=1	455 (76.2%)	307 (77.7%)	0.582	762 (76.8%)
Obesity	81 (34.9%)	115 (43.6%)	0.050	196 (39.5%)
Ischemic cardiomyopathy	255 (42.7%)	154 (39.0%)	0.243	409 (41.2%)
COPD	187 (31.3%)	93 (23.5%)	0.008	280 (28.2%)
Connective tissue disease	163 (27.3%)	88 (22.3%)	0.075	251 (25.3%)
Cerebrovascular disease	161 (27.0%)	68 (17.2%)	<0.001	229 (23.1%)
Mild Liver disease	4 (1.8%)	2 (0.9%)	0.387	6 (1.3%)
Diabetes	241 (40.4%)	162 (41.0%)	0.840	403 (40.6%)
Chronic kidney failure	169 (28.3%)	85 (21.5%)	0.017	254 (25.6%)
Solid tumor	246 (41.2%)	122 (30.9%)	<0.001	368 (37.1%)
Liver failure	134 (22.4%)	58 (14.7%)	0.002	192 (19.4%)
Hematologic disease	16 (7.3%)	9 (3.9%)	0.120	25 (5.6%)
Peptic ulcer disease	8 (3.7%)	5 (2.2%)	0.359	13 (2.9%)
Dementia	176 (29.5%)	74 (18.7%)	<0.001	250 (25.2%)
Arterial hypertension	207 (34.7%)	160 (40.5%)	0.063	367 (37.0%)
Chronic heart failure	36 (16.4%)	30 (13.2%)	0.339	66 (14.7%)
Peripheral vascular disease	50 (22.2%)	56 (24.0%)	0.646	106 (23.1%)
CCI, mean (SD)	7.2 (10.0)	7.6 (10.8)	0.383	7.4 (10.3)
Vaccination status			0.825	
Not vaccinated	191 (32.0%)	120 (30.4%)		311 (31.4%)
One dose	39 (6.5%)	22 (5.6%)		61 (6.1%)
Two doses	360 (60.3%)	249 (63.0%)		609 (61.4%)
Three doses	7 (1.2%)	4 (1.0%)		11 (1.1%)

* Chi-square or Mann–Whitney test as appropriate.

**Table 2 viruses-15-00294-t002:** Vital signs at admission and in follow-up by treatment group.

	Treatment Strategy				
Characteristics	N	Glucocorticoids	Glucocorticoids + Tocilizumab	*p*-Value *	Total
Systolic blood pressure	723			0.823	
Median (IQR)		130 (120, 145)	130 (120, 145)		130 (120, 145)
Diastolic blood pressure	722			0.512	
Median (IQR)		75 (67, 83)	75 (70, 80)		75 (70, 81)
Sofa Score	359			0.019	
Median (IQR)		2 (2, 4)	3 (2, 4)		2 (2, 4)
Baseline PaO_2_/FiO_2_, mmHg	573			<0.001	
Median (IQR)		276 (226, 318)	235 (154, 280)		261 (200, 303)
0–250, *n*(%)		130 (35.8%)	123 (58.6%)	<0.001	253 (44.2%)
0–150, *n*(%)		43 (11.8%)	51 (24.3%)	<0.001	94 (16.4%)
HFNO PaO_2_/FiO_2_, mmHg	228			0.694	
Median (IQR)		102 (65, 158)	103 (72, 153)		103 (71, 153)
^&^ NIV PaO_2_/FiO_2_, mmHg	123			0.304	
Median (IQR)		115 (61, 150)	92 (69, 114)		94 (67, 127)
^&^ IMV PaO_2_/FiO_2_, mmHg	71			0.102	
Median (IQR)		109 (68, 169)	80 (63, 110)		87 (64, 126)
^&^ Respiratory rate	792			<0.001	
Median (IQR)		20 (18, 26)	24 (20, 28)		22 (19, 28)

* Chi-square or Mann–Whitney test as appropriate, ^&^ Baseline values at the time of initiating respiratory support.

**Table 3 viruses-15-00294-t003:** Laboratory parameters by treatment group at baseline.

	Treatment Strategy			
Baseline laboratory parameters	Glucocorticoids	Glucocorticoids + Tocilizumab	*p*-Value *	Total
	N = 421	N = 287		N = 708
Leukocytes, /mm^3^, Median (IQR)	7070 (4740, 9750)	6360 (4870, 9150)	0.126	6740 (4815, 9490)
Neutrophils, /mm^3^, Median (IQR)	5417 (3470, 8015)	5204 (3586, 7932)	0.922	5349 (3538, 8012)
Lymphocytes, /mm^3^, Median (IQR)	938.0 (685.0, 1350)	835.5 (594.0, 1212)	0.011	903.0 (652.0, 1269)
Platelets, 10^3^/mm^3^, Median (IQR)	209.0 (156.0, 264.0)	203.0 (158.0, 258.0)	0.348	207.0 (157.0, 262.5)
Alanine amino-transferase (ALT), U/L, Median (IQR)	26.0 (17.0, 44.0)	29.0 (19.0, 49.0)	0.111	28.0 (18.0, 47.0)
INR, Median (IQR)	1.0 (1.0, 1.1)	1.0 (1.0, 1.1)	<0.001	1.0 (1.0, 1.1)
Creatinine, mg/dL, Median (IQR)	0.9 (0.8, 1.2)	0.9 (0.7, 1.1)	0.208	0.9 (0.7, 1.2)
eGFR, mL/min, Median (IQR)	80.3 (52.3, 95.8)	82.7 (58.9, 94.9)	0.501	81.8 (55.9, 95.6)
60 + mL/min, *n*(%)	296 (70.3%)	213 (74.2%)	0.355	509 (71.9%)
31–60 mL/min, *n*(%)	89 (21.1%)	57 (19.9%)		146 (20.6%)
0–30 mL/min, *n*(%)	36 (8.6%)	17 (5.9%)		53 (7.5%)
C-reactive protein, mg/dL, Median (IQR)	6.0 (2.0, 14.0)	7.0 (4.0, 15.0)	<0.001	6.0 (3.0, 15.0)
IL-6, mg/L, Median (IQR)	27.6 (6.1, 98.5)	175.0 (41.5, 716.4)	<0.001	71.0 (16.2, 301.5)
Procalcitonin, ng/mL, Median (IQR)	0.1 (0.1, 0.4)	0.1 (0.1, 0.3)	0.471	0.1 (0.1, 0.4)
D-dimer, ng/mL, Median (IQR)	945.0 (540.0, 2250)	910.0 (570.0, 1470)	0.375	930.0 (550.0, 1850)
0–500 ng/mL, *n*(%)	89 (21.8%)	52 (18.2%)	0.041	141 (20.3%)
501–4000 ng/mL, *n*(%)	268 (65.7%)	211 (74.0%)		479 (69.1%)
4000 + ng/mL, *n*(%)	51 (12.5%)	22 (7.7%)		73 (10.5%)
Haemoglobin, g/dL	13.3 (11.9, 14.3)	13.7 (12.5, 14.6)	0.004	13.4 (12.1, 14.4)
Lactate dehydrogenase, U/L	525.0 (430.0, 660.0)	622.0 (471.0, 775.0)	<0.001	555.0 (450.0, 713.0)

* Chi-square or Mann–Whitney test.

**Table 4 viruses-15-00294-t004:** Unadjusted and adjusted HR of death from fitting Cox regression models—all participants.

	Hazard Ratios of Death (95% CI)	*p*-Value
**Unadjusted**		
**Glucocorticoids**	1	
**Glucocorticoids + tocilizumab**	0.66 (0.46, 0.95)	0.024
**Adjusted for time-fixed covariates ^1^**		
**Glucocorticoids**	1	
**Glucocorticoids + tocilizumab**	0.55 (0.35, 0.86)	0.009
**Adjusted for time-varying covariates ^2^**		
**Glucocorticoids**	1	
**Glucocorticoids + tocilizumab**	0.64 (0.40, 1.00)	0.048
**Weighted ^3^**		
**Glucocorticoids**	1	
**Glucocorticoids + tocilizumab**	0.59 (0.38, 0.90)	0.015

^1^ standard Cox model adjusted for age, ethnicity, CCI, baseline CRP, and PaO_2_-FiO_2_ ratio, ^2^ standard Cox model adjusted for age, ethnicity, CCI, baseline PaO_2_-FiO_2_ ratio and CRP and time-varying use of remdesivir invasive mechanical ventilation, PaO_2_-FiO_2_ ratio, and CRP, ^3^ weighted Cox model controlled for age, ethnicity, CCI, baseline PaO_2_-FiO_2_ ratio and CRP and time-varying use of remdesivir, invasive mechanical ventilation, PaO_2_-FiO_2_ ratio, and CRP using IPW.

## Data Availability

The datasets generated during and/or analyzed during the current study are not publicly available, but are available from the corresponding author on reasonable request.

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
