# Peer review of "Do All Critically Ill Patients with COVID-19 Disease Benefit from Adding Tocilizumab to Glucocorticoids? A Retrospective Cohort Study"

_viruses, 2023, doi:10.3390/v15020294_

Round 1
Reviewer 1 Report
Please check attachment.

Reviewer 2 Report
At the outset, I congratulate the authors on doing comprehensive work. Your data is large and represents a true picture of the real-world use of tocilizumab. Compiling data from nearly 1000 subjects must have been an exhaustive task, and adjusting for the variables is even more so.
Despite the pros, I am very concerned about the writing, which needs extensive work. The present form of the manuscript is no-where near the quality of the presented work. Even the fonts are different in the same sections; several sentences are incomplete and major sections of the introduction and discussion are just plain overlaps. This reflects poorly on the journal and work of the investigators. (I genuinely apologise if my words seem harsh.)
The scientific part is exceptional though I would suggest the following points.
1. Adverse events must be graded according to CTCAE version 5 and presented in the same format.
2. The two groups do not have similar populations. And hence this can lead to several confounding results, even if the adjustments have been made. This lacuna must be acknowledged and highlighted.
3. Baseline characteristic table is inadvertently and unnecessarily extensive and must be shortened.
4. Adjustments for subgroups of patients on respiratory support like HFNO, NIV, or IMV should be made separately.
5. Another analysis which compares the timing of the use of TCZ should be added (within 10 days of illness onset versus later.).
6. TCZ led to significant improvement in the initial half of the Kaplan-Meier curve, but later, the curves moved nearer. this can be explained in the discussion.
Round 2
Reviewer 2 Report
Thank you for making the desired changes. Fonts are still different in different sections, but I think that would be changed during the proofreading.